# Long term cost-effectiveness analysis of IDegLira in the treatment of type 2 diabetes patients compared to GLP-1RA added to basal insulin after IDegLira entered the national reimbursement drug list in China

**Dunming Xiao[1,2], Junling Weng[1,2], Lei Zhang[3], Chang Xing[4], Yan Wei[1,2]\*, Yingyao Chen [1,2]\***

**1** School of Public Health, Fudan University, Shanghai, China, **2** NHC Key Laboratory of Health Technology Assessment, Fudan University, Shanghai, China, **3** Novo Nordisk (Shanghai) Pharmaceutical Trading Co., Ltd, Shanghai, China, **4** Novo Nordisk (China) Pharmaceuticals Co., Ltd, Beijing, China

\* yanwei@fudan.edu.cn (YW); yychen@shmu.edu.cn (YC)

## Abstract

### Background

Degludec insulin liraglutide injection is the world's first and only approved basal insulin GLP-1RA injection in China (GLP-1RA stands for Glucagon-Like Peptide-1 Receptor Agonist. This study aimed to evaluate the long-term cost-effectiveness of IDegLira versus GLP-1RA added to basal insulin regimen (combined regimen) for patients with type 2 diabetes in China.

### Methods

Based on the perspective of the health system, the study adopts the Swedish Institute for Health Economics diabetes cohort model. The baseline and clinical efficacy data in the model are from the EXTRA study. The cost includes glucose-lowering medication, background treatment cost, and complication treatment costs. The prices of IDegLira and GLP-1RA are based on the national medical insurance payment standards. Insulin adopts the national volume-based procurement average price. Other cost and utility data are sourced from published literature. The health outcome indicator is quality-adjusted life years (QALYs). The simulation time horizon is 30 years. The discounting rate of cost and health outcomes is 5%.

### Results

In clinical output, IDegLira could reduce the cumulative incidence of various chronic complications in patients compared to the combined regimen. The eye diseases (background, proliferative retinopathy, macular edema, and visual loss) decreased by 24.4%, 41.1%, 18.9%, and 12.2%, respectively. Neuropathy decreased by 17.3%. Proteinuria and end-stage renal

**Data Availability Statement:** All relevant data are within the manuscript and its Supporting Information files.

**Funding:** This study was funded by Novo Nordisk (China) Pharmaceuticals Co., Ltd. However, the analysis was independently conducted by the researchers and was not influenced by the company.

**Competing interests:** The authors have declared that no competing interests exist.

disease decreased by 25.9% and 17.6% respectively. Ischemic heart disease, heart failure, and myocardial infarction (stroke) decreased by 0.8%, 1.1%, and 4.7%, respectively. In the base-case analysis, IDegLira compared to the combined regimen shows an incremental cost of -34,254 CNY and an incremental QALYs of 0.436. Under the threshold of 1 times the per capita GDP of China in 2022 (85,698 CNY), IDegLira is a dominant scheme with lower cost and better health outcome. In probabilistic sensitivity analysis, the probability of IDegLira being cost-effective is 100%, indicating that the base-case analysis results are robust.

## Conclusion

Compared with the combined regimen, the use of IDegLira for Chinese patients with type 2 diabetes can improve long-term health output, save medical costs, and is a dominant scheme.

## Introduction

Diabetes Mellitus (DM) is a group of chronic lifelong diseases characterized by high blood glucose levels, and its prevalence, disability, morbidity, mortality, and degree of harm to people's health are second only to malignant tumors, cardiovascular and cerebrovascular diseases, and it is one of the most important chronic non-communicable diseases in the world [1].WHO classifies diabetes mellitus into 4 types based on a etiological evidence [2]: Type 1 Diabetes Mellitus (T1DM), type 2 diabetes mellitus (T2DM), other specific types of diabetes mellitus, and gestational diabetes mellitus. Among the main types of diabetes mellitus, the number of patients with type 2 diabetes mellitus (T2DM) accounts for more than 90% of the total number of patients with diabetes mellitus [3]. The International Diabetes Federation (IDF) officially released the 10th edition of the Global Diabetes Atlas [4], and the number of people with diabetes in China is expected to be 140.9 million and 174.4 million in 2021 and 2045, respectively, which is the highest in the world. In terms of health expenditures, China's adult (20–79 years old) health expenditures on diabetes will be US$165.3 billion in 2021, ranking second globally, bringing a heavy burden of the disease to patients, families, and society.

Pancreatic β-cell dysfunction and insulin resistance are the core mechanisms in the pathogenesis of T2DM, and both involve multiple vital organs and tissues, which together lead to glucose metabolism disorders [3]. If patients with T2DM do not achieve glycemic control based on a combination of lifestyle and oral glucose-lowering medication, insulin therapy should be initiated as early as possible [3]. Basal insulin analogs mimic physiologic basal insulin secretion and reduce β-cell load [5]; glucagon-like peptide-1 receptor agonists (GLP-1RA) can improve pancreatic β-cell function and reduce pancreatic β-cell apoptosis as well as improve insulin resistance [6–8]. The combination of basal insulin and GLP-1RA provides a complementary mechanism to reduce the dose of insulin used and less weight gain in patients [9].

IDegLira (hereinafter referred to as Insulin Degludec and Liraglutide Injection) is the world's first and only reimbursed basal insulin and GLP-1RA injection in China. As a new type of diabetes therapeutic drug, it can act on insulin receptors and GLP-1 receptors at the same time, thus realizing a dual-receptor, complementary mechanism, and entered the national health insurance catalog in 2022.

This study aims to evaluate the long-term cost-effectiveness of Degludec insulin liraglutide injection in adult type 2 diabetes mellitus patients with poor glycemic control, to inform health

care and health policymakers, as well as provide evidence for hospital drug selection and rational clinical use.

## Materials and methods

### Target population

Chinese patients with type 2 diabetes who have poor glycemic control.

### Interventions

The treatment regimen in the intervention group was IDegLira once-daily injection, and the treatment regimen in the control group was GLP-1RA once-daily injection in combination with basal insulin once-daily injection.

### Modelling approach

In this study, we used the Swedish Institute for Health Economics (IHE) diabetes cohort model to simulate the long-term health outputs and costs of treatment with IDegLira in Chinese patients with type 2 diabetes mellitus based on the perspective of China's healthcare system. The IHE diabetes cohort model is a published, validated, and expert-reviewed model [10], which is suitable for the simultaneous comparison of multiple comparison groups. In recent years, the IHE model has been increasingly used for the economic evaluation of several glucose-lowering drugs (semaglutide, liraglutide, IDegLira, etc.) in Sweden, Canada, and other countries [11–14].

The IHE model was constructed based on Visual Basic for Applications (VBA) built into Microsoft® Excel 2013 to simulate the occurrence of diabetes-related complications through constructing Markov sub-models. The IHE model allows for the simultaneous comparison of one intervention and 12 control groups, with an upper limit of the study timeframe of 40 years.

In this study, two parallel Markov chains were used. By inputting parameters such as patients' baseline characteristics, risk factors for complications, clinical trial results, costs of treating diabetes and its complications, and health utility values, researchers can simulate and obtain long-term health and economic outcomes. Model results include the progression of complications in diabetic patients after receiving the appropriate treatment regimen, life years (LYs), quality-adjusted life years (QALYs), as well as the cost of glucose-lowering treatment, the cost of treatment of complications, the total cost of treatment for patients. These output results of the model are used for economic evaluation and the structure of the IHE model is shown in Fig 1.

### Research time frame and discount rate

According to the recommendations of the Chinese Guidelines for Pharmacoeconomic Evaluation for study time horizon and discount rate, in this study, considering that the baseline mean age of diabetic patients was about 61 years, and in order to obtain the impact of the interventions on the lifelong health outputs and costs of diabetic patients, the time horizon of the study was set to 30 years, and a discount rate of 5% was applied to discount the costs and health outputs [15].

### Baseline characteristics

Data on the baseline characteristics of the patients in this study were obtained primarily from the EXTRA study [16]. The EXTRA study is a retrospective, non-interventional, multicenter,

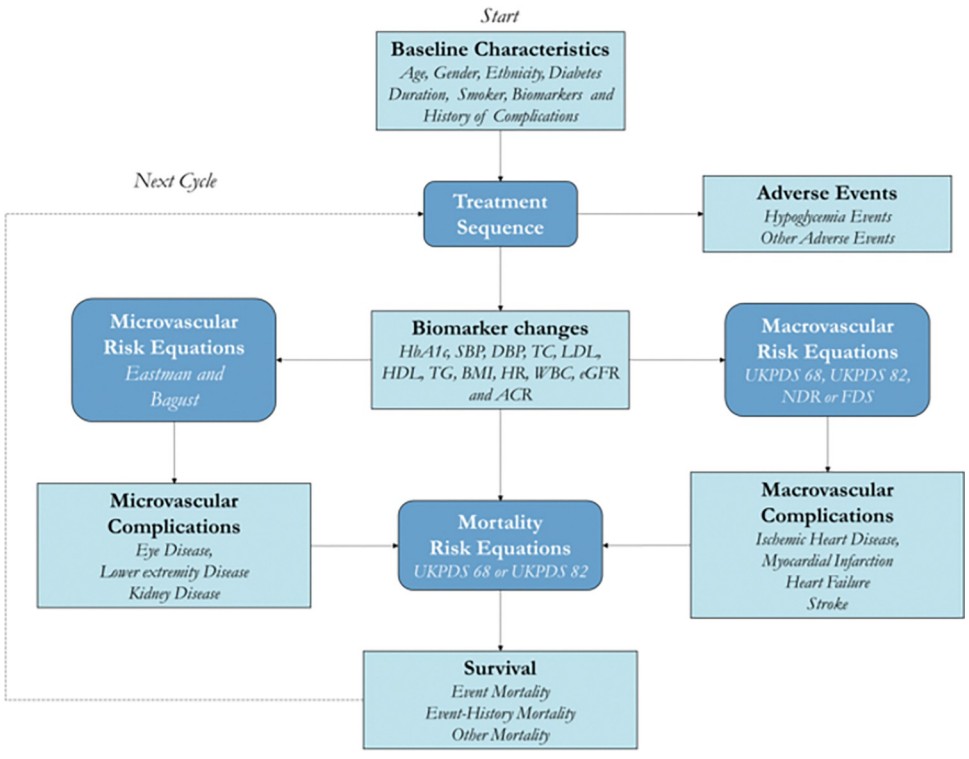

**Fig 1. Structure of the IHE model.**

case analysis study. The study evaluated the efficacy and safety of IDegLira in adults with type 2 diabetes mellitus in a real-world setting. A total of 611 patients from 61 centers in 5 countries were enrolled in the study, and a subgroup of these patients who had switched to IDegLira from a previous GLP-1RA + basal insulin combination regimen was selected for analysis in this study. The mean age of the patients at baseline was 61 years, 35.9% were female, and the mean duration of disease was 14.1 years, as detailed in Table 1.

## Clinical efficacy data

Clinical efficacy data in this study were obtained from the EXTRA study [16], as detailed in Table 2, which showed that treatment with IDegLira significantly reduced HbA$_{1c}$, the incidence of hypoglycemia, and effectively controlled body weight, and systolic blood pressure when compared with the GLP-1RA added to basal insulin regimen. The effects of IDegLira on clinical risk factors reported in the study results were used as the clinical efficacy and compared with patients continuing with GLP-1RA plus basal insulin (assuming no further change in risk factors while on the same therapy). When the subgroup data about GLP-1RA plus basal insulin were not available for the efficacy data, the results of the total population data were used. The Institutional Review Board (IRB) was not applicable for this study because the related data was collected from literatures and website.

Patients who were prescribed IDegLira or combination therapy continued this treatment until their HbA1c levels rose above 8.5%. At that point, they were switched to the intensification basal-bolus regimen with insulin glargine plus three times daily insulin apart. After switching to the intensification insulin regimen, the effect of HbA1c, BMI, hypoglycemic event rates, and annual costs were the same in both treatment arms. Alternative approaches to

**Table 1. Baseline characteristics of diabetic patients in the model.**

| Variant | Mean | Reference |
|---|---|---|
| Demographic characteristics | | |
| Age (years) | 61 | EXTRA [16] |
| Duration of diabetes (years) | 14.1 | EXTRA [16] |
| Proportion of women (%) | 35.90% | EXTRA [16] |
| Proportion of smokers (%) | 26.50% | Li C, et al [17] |
| Baseline risk factors | | |
| Glycated hemoglobin (HbA$_{1c}$, %) | 8.3 | EXTRA [16] |
| Systolic blood pressure (SBP, mmHg) | 141.5 | EXTRA [16] |
| Diastolic blood pressure (DBP, mmHg) | 82.4 | EXTRA [16] |
| Total cholesterol (TC, mmol/L) | 4.82 | EXTRA [16] |
| Low-density lipoprotein cholesterol (LDL, mmol/L) | 2.75 | EXTRA [16] |
| High-density lipoprotein cholesterol (HDL, mmol/L) | 1.21 | EXTRA [16] |
| Triglycerides (TG, mmol/L) | 2.42 | EXTRA [16] |
| Body mass index (BMI, kg/m) $^2$ | 35.9 | EXTRA [16] |
| Heart Rate (HR, bpm) | 72.0 | UKPDS [18] |
| White blood cell count (WBC, 1x10) $^6$ | 6.8 | UKPDS [18] |
| Glomerular filtration rate (eGFR, mL/min/1.73m) $^2$ | 77.5 | UKPDS [18] |

Note: According to the requirements of model input parameter units, some clinical utility values were converted, HbA$_{1c}$ was converted from mmol/mol in clinical studies to % [19] in model input requirements; total cholesterol (TC), low-density lipoprotein cholesterol (LDL), high-density lipoprotein cholesterol (HDL), and triglyceride (TG) were converted from mg/dL to mmol/L in model input requirements. The conversion coefficients were based on the Chinese Guidelines for the Management of Type 2 Diabetes (2020 Edition), i.e., TC, HDL, and LDL were divided by 38.67, and TG divided by 88.57.

treatment switching and long-term parameter progression were evaluated in sensitivity analyses [20]. The trend of HbA1c over time is shown in S1 Fig of the supplementary materials.

## Cost data

In this study, the costs of different treatment options were calculated from the perspective of China's health system, and the cost categories were direct medical costs, including the costs of diabetes glucose-lowering medications, the costs of injectable needles, the costs of self-monitoring of blood glucose (SMBG), and the costs of treating microvascular complications and

**Table 2. Clinical efficacy data.**

| Efficacy indicators | IDegLira | GLP-1RA+basal insulin | Difference between two groups (95% CI) |
|---|---|---|---|
| HbA$_{1c}$ (%) | 7.6 | 8.3 | -0.6 (-0.8; -0.4) p<0.0001 |
| BMI (kg/m$^2$) | 35.83 | 35.9 | -0.07 p>0.05 |
| Systolic blood pressure (mmHg) | 139.7 | 143.2 | -3.5 (-5.7; -1.3) p = 0.0017 |
| Diastolic blood pressure (mmHg) | 83.9 | 83 | 0.9 (-0.4;2.2) p = 0.1617 |
| Hypoglycemia (events/exposed patient-years) | 0.06 | 0.061 | RR 0.98 (0.63;1.54) p = 0.94 |

Note: BMI Body mass index, HbA1c glycated hemoglobin, RR rate ratio, CI confidence interval. Systolic blood pressure and diastolic blood pressure use the results of the total population data; because only the incidence of hypoglycemia between all treatment regimens before switching and after switching to IDegLira was reported in EXTRA, the incidence of hypoglycemia in the total population of IDegLira was used in this study in the context of the incidence of hypoglycemia in the total population of IDegLira in EXTRA regarding the difference in the rate of hypoglycemia between liraglutide basal insulin (insulin glargine or detemir) and IDegLira RR for non-severe hypoglycemia in the indirect comparison study [21], to obtain the incidence of hypoglycemia in the reference regimen group.

**Table 3. Costs associated with glucose-lowering therapy for diabetes in the model.**

| Cost of treatment regimens | IDegLira | GLP-1RA + basal insulin |
|---|---|---|
| Cost of medicines (CNY/day) | 26.8 | GLP-1RA 25.8; basal insulin 8.2 |
| Cost of needles (CNY/day) | 2.6 | 5.1 |
| SMBG costs (CNY/day) | 8.2 | 8.2 |
| Total cost (CNY/year) | 13,702 | 17,277 |

macrovascular complications. Specific cost data are detailed in Tables 3 and 4 where the cost of glucose-lowering medicines is calculated based on the unit price and dosage of the medicine. The price of IDegLira adopts the national health insurance payment standard; the price of insulin in the GLP-1RA+basal insulin combination treatment group adopts the national volume-based procurement standard, and the price of GLP-1RA adopts the national health insurance payment standard or the national average winning price in 2023(details see S1 and S2 Tables of supplementary materials). The proportion of patients using different drugs of GLP-1RA was obtained from the EXTRA study, and the drug cost was calculated by the weighted average of the proportion of use of each drug; the basal insulin dosage was obtained from the EXTRA study, and the basal insulin drug cost was calculated by adopting the average winning bid price and average dosage.

**Table 4. Cost of treating diabetic complications in the model (CNY/year).**

| Treatment cost categories | Event cost | State cost | Reference |
|---|---|---|---|
| Background treatment* | 3,349 | / | Xie, et al [22] |
| Background diabetic retinopathy | 15,757 | 907 | He, et al [24] |
| Proliferative diabetic retinopathy | 15,757 | 907 | He, et al [24] |
| Macular edema | 16,444 | 2,163 | Wu, et al [25] |
| Proliferative diabetic retinopathy and macular edema | 32,201 | 3,070 | hypothetical value** |
| Severe vision loss | 14,950 | 11,538 | Deng, et al [26] |
| Symptomatic neuropathy | 18,768 | 7,201 | Wu, et al [25] |
| Peripheral vascular disease | 28,220 | 10,968 | Wu, et al [25] |
| Amputation of the lower limbs | 22,625 | 18,035 | Deng, et al [26] |
| Massive albuminuria | 14,911 | 5,668 | Duan, et al [27] |
| End-stage renal disease | 158,396 | 127,391 | Duan, et al [27] |
| | | | Su, et al [28] |
| Ischemic heart disease | 48,446 | 8,649 | Deng et.al [26] |
| Myocardial infarction | 80,620 | 25,485 | Duan, et al [27] |
| | | | Su, et al [28] |
| Cerebral hemorrhage | 31,923 | 15,793 | Duan, et al [27] |
| | | | Su, et al [28] |
| Heart failure | 38,623 | 20,489 | Duan, et al [27] |
| | | | Su, et al [28] |
| Non-severe hypoglycemic events | 880 | 0 | Duan, et al [27] |
| | | | Su, et al [28] |
| Severe hypoglycemic events | 13,790 | 0 | Duan, et al [27] |
| | | | Su, et al [28] |

*: consists of Education and consultation, Visiting doctors, Monitoring blood glucose, Screening for foot diseases/ eye diseases/ microalbuminuria and so on.

**: Costs for proliferative diabetic retinopathy and macular edema are assumed to be the cumulative cost of treatment for proliferative diabetic retinopathy and the cost of treatment for macular edema. Event cost is the direct medical cost in the year of the event. State cost is the annual cost of maintaining treatment each year, including the year of the event.

Injection needle costs were calculated based on the number of injections and needle prices in the two regimens, and needle prices were calculated using the average national winning price of 32g needles. SMBG costs and diabetes complications treatment costs were derived from data in published literature and were adjusted to the 2022 level by the China Medical Consumer Price Index [22–24]. For detailed information please check the supplement.

## Utility value data

The utility value data used in the study are detailed in Table 5. The utility data related to diabetes-related health status and demographic characteristics were mainly obtained from the quality of life in Chinese patients with type 2 diabetes Study, and the unreported data were used from published literature data mainly from Asian populations.

## Results

### Cumulative incidence of chronic complications of diabetes mellitus in long-term simulations

The cumulative incidence of chronic complications of diabetes for the long-term simulation is shown in Table 6, and the cumulative incidence of chronic complications was lower in the IDegLira than in the GLP-1RA + basal insulin combination group.

**Table 5. Utility value data.**

| Variant | Mean | SE | Reference |
|---|---|---|---|
| Baseline | 0.936 | 0.120 | Li, et al [17] |
| Background diabetic retinopathy | -0.023 | 0.002 | Mok, et al [29] |
| Proliferative diabetic retinopathy | -0.023 | 0.002 | Mok, et al [29] |
| Macular edema | -0.019 | 0.002 | Li, et al [17] |
| Proliferative diabetic retinopathy and macular edema | -0.023 | 0.002 | Mok, et al [29] |
| Severe vision loss | -0.049 | 0.005 | Li, et al [17] |
| Symptomatic neuropathy | -0.026 | 0.003 | Li, et al [17] |
| Peripheral vascular disease | -0.032 | 0.003 | Li, et al [17] |
| Amputation of the lower limbs | -0.139 | 0.014 | Li, et al [17] |
| History of lower extremity amputations | -0.139 | 0.014 | Li, et al [17] |
| Massive albuminuria | -0.030 | 0.003 | Li, et al [17] |
| End-stage renal disease | -0.092 | 0.009 | Li, et al [17] |
| Ischemic heart disease | -0.068 | 0.007 | Li, et al [17] |
| Myocardial infarction events | -0.050 | 0.005 | Li, et al [17] |
| History of myocardial infarction | -0.050 | 0.005 | Li, et al [17] |
| Subsequent myocardial infarction events | -0.012 | 0.001 | CADTH Report [30] |
| History of subsequent myocardial infarction | -0.012 | 0.001 | CADTH Report [30] |
| Stroke event | -0.106 | 0.011 | Li, et al [17] |
| History of stroke | -0.106 | 0.011 | Li, et al [17] |
| Subsequent stroke events | -0.040 | 0.004 | CADTH Report [30] |
| History of subsequent stroke events | -0.040 | 0.004 | CADTH Report [30] |
| Heart failure | -0.186 | 0.019 | Li, et al [17] |
| Age (for each additional 10 years) | -0.024 | 0.002 | Bagust, et al [31] |
| Females | -0.012 | 0.001 | Li, et al [17] |
| Duration of diabetes mellitus (per 10-year increment) | -0.016 | 0.002 | Johansen, et al [11] |
| Overweight (per 1kg/m increase in BMI) [2] | -0.006 | 0.001 | Beaudet, et al [32] |
| Non-severe hypoglycemic events | -0.014 | 0.0014 | CADTH Report [30] |
| Severe hypoglycemic events | -0.047 | 0.0047 | CADTH Report [30] |
| 1 injection per day | -0.00805 | 0.0008 | McEwan, et al [33] |
| 2 injections per day | -0.0101 | 0.0010 | McEwan, et al [33] |

**Table 6. Cumulative incidence of chronic complications of diabetes mellitus in long-term simulations.**

| Variant | IDegLira | GLP-1RA+basal insulin | Relative Risk |
|---|---|---|---|
| Mortality | | | |
| Cumulative Mortality | 88.89% | 89.59% | 0.99 |
| Mortality from Cardiovascular Events (Myocardial Infarction and Stroke) | 17.41% | 18.27% | 0.95 |
| Cumulative Incidence of Microvascular Complications | | | |
| Eye Complications | | | |
| Background Retinopathy | 26.32% | 34.81% | 0.76 |
| Proliferative Retinopathy | 2.27% | 3.86% | 0.59 |
| Macular Edema | 14.63% | 18.04% | 0.81 |
| Proliferative Retinopathy and Macular Edema | 2.03% | 3.38% | 0.60 |
| Severe Vision Loss | 4.57% | 5.20% | 0.88 |
| Lower Extremity Disease | | | |
| Symptomatic Neuropathy | 4.23% | 5.11% | 0.83 |
| Peripheral Vascular Disease | 16.16% | 15.97% | 1.01 |
| Lower Extremity Amputation | 38.79% | 39.13% | 0.99 |
| Kidney Disease | | | |
| Microalbuminuria | 20.51% | 22.58% | 0.91 |
| Macroalbuminuria | 10.41% | 14.05% | 0.74 |
| End-stage Renal Disease | 5.92% | 7.19% | 0.82 |
| Macrovascular Complications | | | |
| Ischemic Heart Disease | 15.56% | 15.68% | 0.99 |
| Myocardial Infarction | | | |
| First Myocardial Infarction | 20.59% | 21.43% | 0.96 |
| Subsequent Myocardial Infarction | 3.35% | 3.40% | 0.98 |
| Stroke | | | |
| First stroke | 8.92% | 9.67% | 0.92 |
| Subsequent Stroke | 3.35% | 3.47% | 0.96 |
| Heart Failure | 9.09% | 9.18% | 0.99 |
| Cardiovascular Disease (Myocardial Infarction & Stroke) | 26.98% | 28.32% | 0.95 |

## Long-term simulated costs of treatment related to chronic complications of diabetes mellitus

Long-term simulated costs of treatment related to chronic complications of diabetes are shown in Table 7, and except for costs of background treatment and lower extremity disease treatment, costs of treatment related to chronic complications were lower in the IDegLira than in the GLP-1RA+basal insulin group.

## Results of base case analysis

Compared to the GLP-1RA + basal insulin combination regimen, IDegLira increased quality-adjusted life years (QALYs) by 0.436 QALYs, resulting in a total direct healthcare cost savings of 34,254 CNY, making it a cost-effective regimen, as shown in Table 8.

## Results of one-way sensitivity analysis

In this study, a one-way sensitivity analysis was conducted for the key parameters, in which the cost and QALY discount rate were floated from 0% to 8%, and the remaining parameters were used to float up and down by 10%. The results of the one-way sensitivity analysis showed

**Table 7. Costs of treatment associated with chronic complications of diabetes simulated over time.**

| Costs | IDegLira | GLP-1RA + basal insulin | Increment |
|---|---|---|---|
| Background treatment costs | 36,254 | 36,109 | 145 |
| Diabetes treatment | 144,820 | 149,211 | -4,390 |
| Hypoglycemia | 48,052 | 73,027 | -24,975 |
| Eye diseases | 8,833 | 10,293 | -1,459 |
| Lower extremity disease | 80,919 | 80,794 | 126 |
| Kidney disease | 25,679 | 28,285 | -2,606 |
| Ischemic heart disease | 12,646 | 12,687 | -41 |
| Myocardial infarction | 29,393 | 29,930 | -537 |
| Stroke | 12,172 | 12,560 | -388 |
| Heart failure | 13,724 | 13,854 | -129 |
| Total cost | 412,494 | 446,748 | -34,254 |

that the top three parameters that had the greatest impact on the results were: cost discount rate, cost of IDegLira and QALYs discount rate, as detailed in Fig 2.

## Probabilistic sensitivity analysis

Probabilistic sensitivity analysis was performed by using a second order Monte Carlo approach with 1000 sample simulations and the results of the analysis are presented in an incremental cost-effect scatterplot and a plot of acceptable cost-effectiveness curves in Figs 3 and 4. As can be seen in the scatterplot, all scatters are in the fourth quadrant indicating that the probability of IDegLira being cost-effective is higher than that of the GLP-1RA + basal insulin regimen. Further, the cost-effectiveness acceptability curves show that the probability of IDegLira being cost-effective compared to the GLP-1RA + basal insulin regimen is consistently 100%, indicating that the results of the base case analysis are robust.

## Discussion

This is the first study to use the IHE diabetes cohort model to simulate the long-term health outputs and costs of IDegLira treatment for Chinese patients with type 2 diabetes after IDegLira entered the national health insurance catalog. It was found that better control of HbA$_{1c}$ by IDegLira and its better safety profile resulted in a lower cumulative incidence of diabetic complications in patients in the IDegLira group than in the GLP-1RA combination with basal insulin group. In terms of patients' lifetime (30-year) health outputs and costs, patients in the IDegLira group experienced an increase in QALYs of 0.436 compared to the GLP-1RA + basal insulin group, along with a reduction in total direct healthcare costs of 342545 CNY, which

**Table 8. Results of base case.**

| Variant | IDegLira | GLP-1RA + basal | Increment |
|---|---|---|---|
| Life expectancy (years) | 10.824 | 10.781 | 0.043 |
| Quality-adjusted life years (QALYs) | 5.685 | 5.249 | 0.436 |
| Total direct medical costs (CNY) | 412,494 | 446,748 | -34,254 |
| Treatment costs | 229,127 | 258,346 | -29,220 |
| Cost of treating microvascular complications | 115,431 | 119,371 | -3,940 |
| Cost of treating macrovascular complications | 67,936 | 69,031 | -1,095 |
| Incremental Cost Effectiveness Ratio (ICER, CNY/QALY) | | | Dominant |

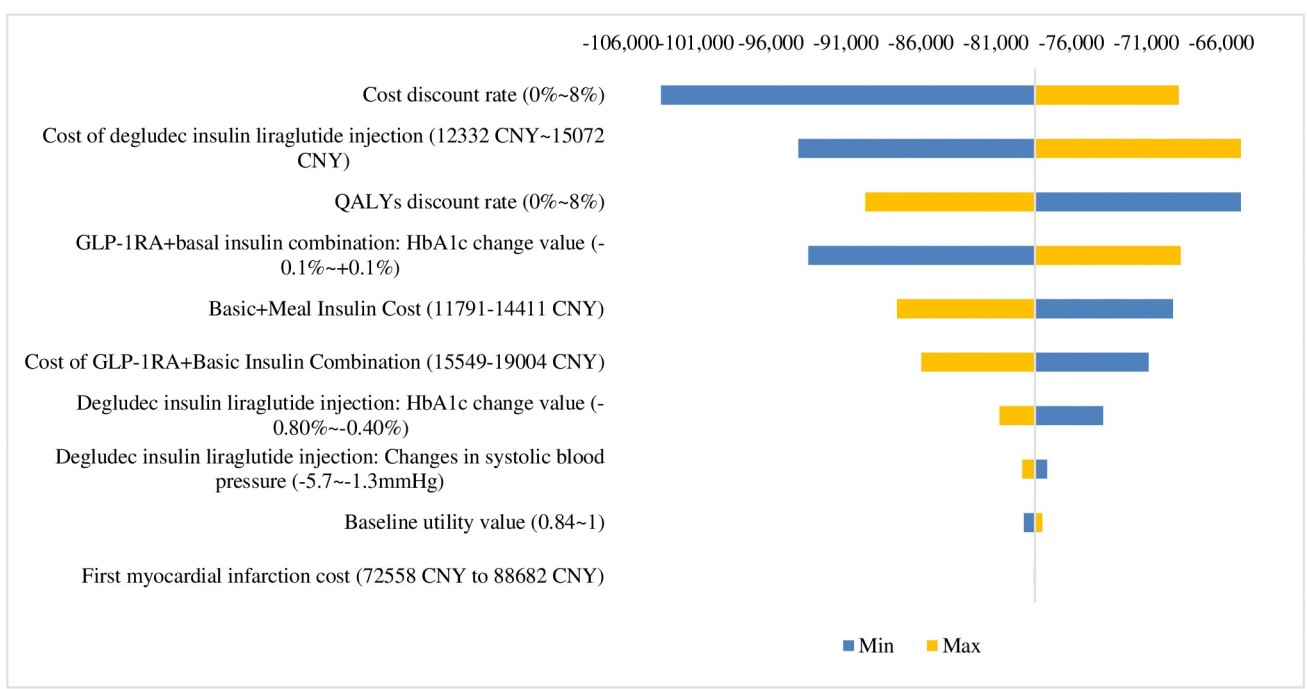

**Fig 2. One-way sensitivity analysis tornado diagram (ICER).**

indicates IDegLira is dominant in China and probabilistic sensitivity analyses showed robust results. To study the impact of changes in baseline characteristics on the results, we conducted univariate sensitivity analyses on parameters including age, proportion of females, duration of diabetes, smoking rate, HbA1c, systolic blood pressure, and total cholesterol, within the ranges relevant to the Chinese population. The results showed that these factors did not rank within the top 10 influential factors in the tornado plot, indicating minimal impact on the results. We also applied the Consolidated Health Economic Evaluation Reporting Standards (CHEERS) checklist [34] to ensure thorough and transparent reporting of our economic evaluation. This

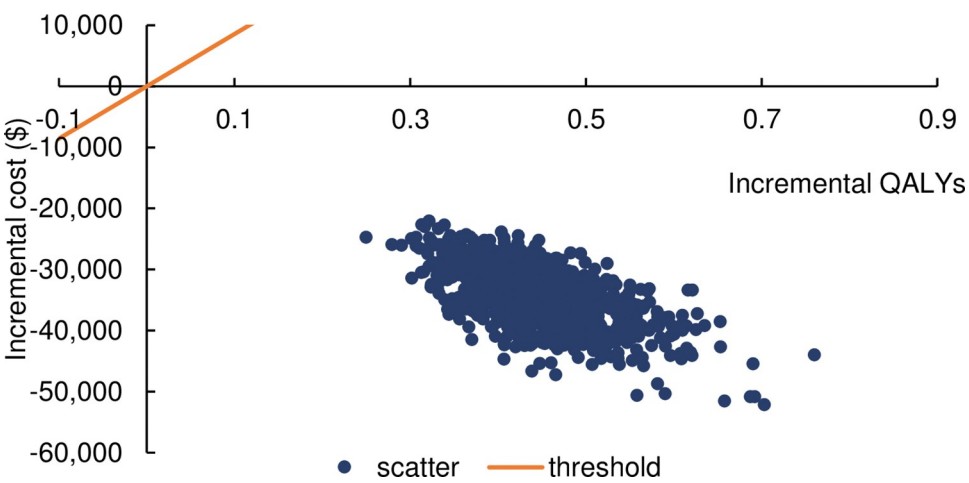

**Fig 3. Probabilistic analysis scatterplot.**

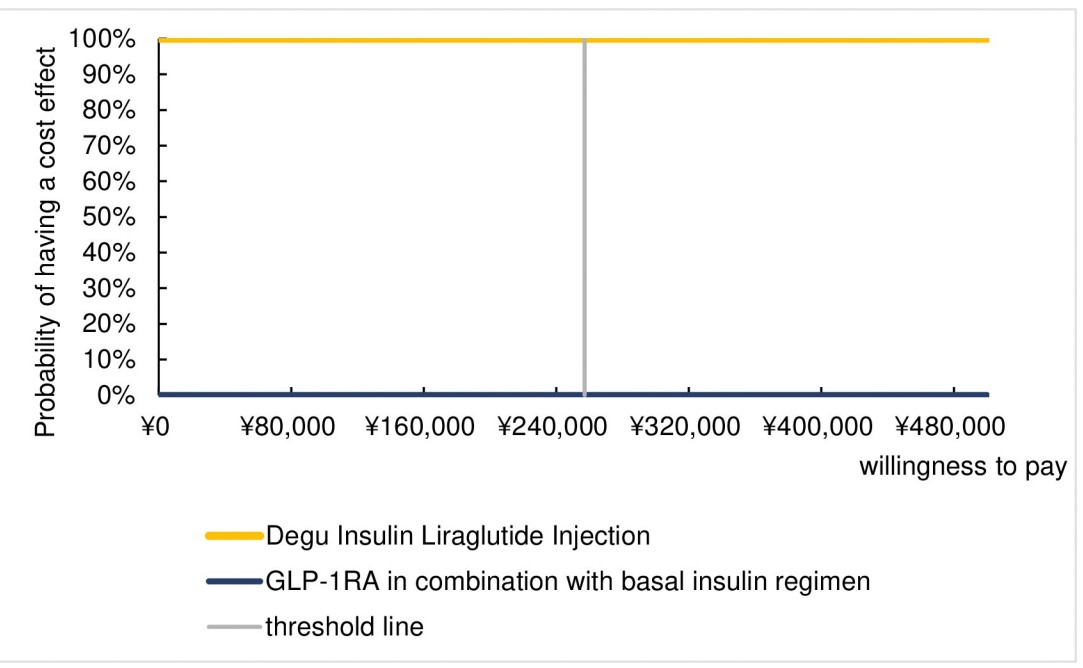

**Fig 4. Cost-effectiveness acceptable curve.**

framework guided our methodology and reporting practices (details see S3 Table of supplementary materials).

In addition, the results of this study are generally consistent with cost-effectiveness studies conducted overseas. The cost-effectiveness studies conducted in the United States [35], the United Kingdom [36], Sweden [13], and Spain [37] showed that IDegLira was dominant (with lower costs and better outcomes) compared with GLP-1RA in combination with basal insulin. The Czech [38] study showed that IDegLira is a cost-effective treatment, improving outcomes with higher costs and the ICER is under the willingness to pay threshold compared to GLP-1RA in combination with basal insulin.

At the same time, there are some limitations of this study. First, the clinical data of this study derived from a real-world single-arm study of patients with type 2 diabetes mellitus (EXTRA study [16]), and although the sample size of the study was large, including 611 patients in 61 centers in 5 countries, there is a lack of direct evidence on whether the differences in drug effects, as manifested in foreign patients, apply to Chinese patients. Due to the lack of Chinese population studies of IDegLira versus GLP-1RA in combination with basal insulin, based on the recommendation of the China Pharmacoeconomic Evaluation Guidelines 2020, clinical effect data from other countries or regions can be used if clinical data from local patients are lacking [10]. Secondly, the IHE diabetes cohort model uses the UKPDS 82 risk prediction equation for the probability of mortality risk, which is not based on the Chinese population, and whether its prediction of long-term diabetic complications differs from the clinical and epidemiological characteristics of Chinese diabetic patients needs to be further explored. Third, this study was conducted from the perspective of the Chinese health system, and only direct medical costs were considered in the model, not direct non-medical costs and indirect costs. Future research could be conducted from a societal perspective to explore the possible impact of different treatment options on society-wide costs.

In summary, the long-term simulation results of this study using the IHE diabetes cohort model based on real-world effectiveness data of the drug and the latest drug price data in

China showed that the use of IDegLira in Chinese patients with type 2 diabetes resulted in longer life-years and quality-adjusted life-years with lower direct healthcare costs compared with the use of GLP-1RA combined with basal insulin, making it a cost-effective treatment option.

## Supporting information

**S1 Fig. Trend of HbA1c over time.**
(TIF)

**S1 Table. GLP-1RA cost.**
(DOCX)

**S2 Table. Basal insulins cost.**
(DOCX)

**S3 Table. Consolidated Health Economic Evaluation Reporting Standards (CHEERS).**
(XLSX)

## Acknowledgments

I certify that no individuals other than the listed co-authors contributed to this publication.

## Author Contributions

**Conceptualization:** Dunming Xiao, Yan Wei, Yingyao Chen.

**Data curation:** Dunming Xiao, Junling Weng.

**Formal analysis:** Dunming Xiao, Junling Weng.

**Funding acquisition:** Lei Zhang, Chang Xing.

**Methodology:** Dunming Xiao, Yingyao Chen.

**Project administration:** Dunming Xiao, Yan Wei, Yingyao Chen.

**Resources:** Dunming Xiao, Junling Weng.

**Software:** Dunming Xiao.

**Supervision:** Yingyao Chen.

**Validation:** Dunming Xiao.

**Visualization:** Dunming Xiao.

**Writing – original draft:** Dunming Xiao.

**Writing – review & editing:** Lei Zhang, Chang Xing, Yan Wei, Yingyao Chen.

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
