## [Decision Letter · Decision Letter 0]

30 Jul 2024

Long Term Cost-effectiveness Analysis of IDegLira in the Treatment of Type 2 Diabetes Patients Compared to GLP-1RA Added to Basal Insulin after IDegLira Entered the National Reimbursement Drug List in China

PONE-D-24-13585R1

Dear Dr. Chen,

We’re pleased to inform you that your manuscript has been judged scientifically suitable for publication and will be formally accepted for publication once it meets all outstanding technical requirements.

Kind regards,

Timotius Ivan Hariyanto, M.D.

Academic Editor

PLOS ONE

Additional Editor Comments (optional):

Reviewers' comments:

Reviewer's Responses to Questions

**Comments to the Author**

1. If the authors have adequately addressed your comments raised in a previous round of review and you feel that this manuscript is now acceptable for publication, you may indicate that here to bypass the “Comments to the Author” section, enter your conflict of interest statement in the “Confidential to Editor” section, and submit your "Accept" recommendation.

Reviewer #1: All comments have been addressed

Reviewer #2: All comments have been addressed

2. Is the manuscript technically sound, and do the data support the conclusions?

Reviewer #1: Yes

Reviewer #2: Yes

3. Has the statistical analysis been performed appropriately and rigorously? 

Reviewer #1: Yes

Reviewer #2: Yes

4. Have the authors made all data underlying the findings in their manuscript fully available?

Reviewer #1: Yes

Reviewer #2: Yes

5. Is the manuscript presented in an intelligible fashion and written in standard English?

Reviewer #1: Yes

Reviewer #2: Yes

6. Review Comments to the Author

Reviewer #1: (No Response)

Reviewer #2: (No Response)

7. PLOS authors have the option to publish the peer review history of their article (what does this mean?). If published, this will include your full peer review and any attached files.

Reviewer #1: No

Reviewer #2: No

---

## [Author Response · Author response to Decision Letter 0]

21 Aug 2024

Dear editors and reviewers,

Thank you for your comments and suggestions, which have been very helpful in improving the quality of the manuscript. We have responded to each of the reviewers' comments one by one as follows..

Reviewer #1: This study used the Swedish Institute for Health Economics Diabetes Cohort Model to estimate the long-term cost-effectiveness of IDegLira versus GLP-1RA added to basal insulin regimen (combined regimen) for patients with type 2 diabetes in China, based on real-world effectiveness data. Overall, the study covers all aspects of the methodology for economic evaluation and shows that the use of IDegLira is a cost-effective treatment option in Chinese patients with type 2 diabetes. It is a well-designed study that adds new information to the literature.

1.As the authors stated in the manuscript, the analyses were conducted from the perspective of the Chinese healthcare system. The baseline population and treatment effects used in the simulation analyses were sourced from the EXTRA study. Does the EXTRA study include Chinese or Asian patients? In other words, could this analysis be justified to present the study results from the Chinese healthcare system perspective?

Reply：Thank you very much for your suggestions. IDegLira has an RCT (DUAL II China) involving the Chinese population, but the control group was not a combined regimen (GLP-1+basal insulin). Therefore, after conducting a literature search, we ultimately selected the EXTRA study. And this is the health economics research after the launch of IDegLira, so we chose this real-world study. The baseline characteristics of the EXTRA study are comparable to those of Chinese type 2 diabetes patients, such as HbA1c (8.94% vs. 8.3%), age (54.7 vs. 61), and proportion of women (39.5% vs. 35.90%), with relatively minor differences. To study the impact of changes in baseline characteristics on the results, we conducted univariate sensitivity analyses on parameters including age, proportion of females, duration of diabetes, smoking rate, HbA1c, systolic blood pressure, and total cholesterol, within the ranges relevant to the Chinese population. The results showed that these factors did not rank within the top 10 influential factors in the tornado plot, indicating minimal impact on the results.

2.What is the sample size of the subgroup receiving GLP-1RA + basal insulin combination regimen at baseline in the EXTRA study?

Reply：The sample size is 611.

3. Is it reasonable to assume no further change in risk factors for the comparator (GLP-1RA+basal insulin group)?

Reply：The risk factors for the comparator could change in every year. The IHE model incorporates a drift equation for risk factors and a risk prediction equation for complications. Each year, the risk factors are adjusted according to the drift equation, which will help to calculate the annual incidence rate of complications.

4. How is the progression of biomarkers or risk factors in the long-term simulation, such as HbA1c, BMI, and blood pressure?

Reply：The IHE model incorporates a drift equation for these risk factors. Each year, the risk factors are adjusted according to the drift equation.

5.Whether there is combined use of multiple anti-diabetes agents in the EXTRA study, as it is common in real-world practice for the treatment of type 2 diabetes? Did the authors consider the cost of combined anti-diabetes drugs (such as metformin) when calculating the costs associated with glucose-lowering therapy for diabetes in the model (Table 3)?

Reply：There are some oral antidiabetic drugs in the EXTRA study. We did not consider the cost of oral antidiabetic drugs. Because the difference of oral antidiabetic drug use between the two groups is minimal, incremental analysis can effectively offset this difference.

6.What are the basal insulins and GLP-1 RAs that are used in the glucose-lowering therapy cost calculation?

Reply：We calculated the average GLP-1 RAs cost for these 5 medications: Exenatide, Exenatide once-weekly formulation, Lixisenatide, Dulaglutide, and Liraglutide. We calculated the average basal insulins cost for these 4 medications: Insulin deludes, Insulin detemir, Regular insulin U100, and Regular insulin U300. The details will be uploaded as supplementary materials.

7.The CHEERS checklist should be provided alongside this cost-effectiveness analysis.

Reply：OK. Many thanks! We will upload it as supplementary materials.

8.According to the base-case result, the cost savings of IDegLira were mainly due to the reduction in hypoglycemia cost. However, the difference in hypoglycemic incidence between the two groups was small (0.06 vs 0.061), so how should this result be interpreted?

Reply：The difference (0.06 vs 0.061) was the efficacy data in the model for the first year. The model simulated 30 years. Moreover, the therapeutic effects of the two groups are different, and the degree of HbA1c control in the subsequent period is also different, which can lead to differences in hypoglycemic events later. After 30 years of accumulation, the cost difference of these hypoglycemic events will be significant.

9.Were the baseline population characteristics closed to the type 2 diabetes population in China? A sensitivity analysis might be needed to include the baseline characteristics of the T2D Chinese population. 

Reply：The baseline characteristics of the EXTRA study are comparable to those of Chinese type 2 diabetes patients (DUAL II China), such as HbA1c (8.94% vs. 8.3%), age (54.7 vs. 61), and proportion of women (39.5% vs. 35.90%), with relatively minor differences. 

To study the impact of changes in baseline characteristics on the results, we conducted univariate sensitivity analyses on parameters including age, proportion of females, duration of diabetes, smoking rate, HbA1c, systolic blood pressure, and total cholesterol, within the ranges relevant to the Chinese population. The results showed that these factors did not rank within the top 10 influential factors in the tornado plot, indicating minimal impact on the results.

10. Figure 2: It should be ‘insulin degludec’ instead of ‘Degu insulin’, please check.

Reply：OK, many thanks! We will correct it in the Figure 2

Reviewer #2: This study compares the cost-effectiveness of using IDegLira versus GLP-1RA plus basal insulin in the Chinese T2D population. It is a well-written manuscript by a renowned research team from China with a long-standing portfolio in health technology assessment. The research was conducted following standard procedures and rigor, and the outcomes are reasonable. I do not have major concerns about the study. However, I have made a series of minor comments to help the authors improve the transparency of their writing

Minor: 1. The title is a bit confusing. I was initially thinking that the author was comparing IDegLira with GLP-1RA when both served as addon therapies to basal insulin. Please reframe the title to improve the clarity.

Reply：Thank you very much for your suggestions. The new title is “Long Term Cost-effectiveness Analysis of IDegLira in the Treatment of Type 2 Diabetes Patients Compared to GLP-1RA Added to Basal Insulin after IDegLira Entered the National Reimbursement Drug List in China”.

2. abstract: “The cost includes hypoglycemic treatment”. Recommend changing it to “glucose-lowering medication”.

Reply：OK, many thanks!

3. abstract: what is disease management cost? Please make it more specific. What disease? What cost? Drugs?

Reply：The disease management cost means background treatment cost in the table 4. We have changed it to background treatment cost in the abstract. The management cost derived directly from the literature (Ref.22), which consists of education and consultation, visiting doctors, monitoring blood glucose, screening for foot diseases/ eye diseases/ microalbuminuria and so on. The detail drugs were not reported in the literature (Ref.22). We have added explanations below Table 4.

4. abstract: reporting the total cost is not necessary, because your cost was discounted, making it challenging to interpret. Reporting differences in cost between the two arms is much more intuitive, which is missing from the current abstract.

Reply：OK, many thanks! We have modified the abstract.

5. abstract: The willingness to pay threshold used in CEA must be presented in the abstract.

Reply：OK, many thanks! We have modified the abstract.

6. manuscript: please change the term “hypoglycemic agents” to glucose-lowering medication consistently across the manuscript.

Reply：OK, many thanks! We have modified the manuscript.

7. Manuscript: “Clinical efficacy data in this study were obtained from the EXTRA study, as detailed in Table 2, which showed that treatment with IDegLira significantly reduced HbA1c, the incidence of hypoglycemia, and effectively controlled body weight, and systolic blood pressure when compared with the control treatment regimen.” Please clarify what do you mean by “control treatment regimen.

Reply：The control treatment regimen means GLP-1RA added to basal insulin regimen (combined regimen). We have clarified it in the manuscript.

8. “Alternative approaches to treatment switching and long-term parameter progression were evaluated in sensitivity analyses” what are the alternatives? Please spell out.

Reply：The alternatives were insulin glargine combined with three times daily insulin. When the HbA1c reached to 8.5%, the current treatment will be changed to the alternatives. We have modified it in the manuscript.

9. Manuscript: please provide a figure on HbA1c progression overtime. This will help the readers to understand how treatment was switched/escalated during the study period.

Reply：OK. We have added HbA1c progression figure in supplementary materials.

10. Please discuss how different the EXTRA population is from the general T2D insulin users among the Chinese population. It is fine to be different but there must be some discussion to let the reader understand the potential biases.

Reply：We conducted univariate sensitivity analyses on parameters including age, proportion of females, duration of diabetes, smoking rate (%), HbA1c, systolic blood pressure, and total cholesterol, within the ranges relevant to the Chinese population. These analyses showed that these factors did not rank within the top 10 influential factors in the tornado plot, indicating minimal impact on the results. We will provide additional explanations in the discussion section.

11. Please add a CHEERS Checklist.

Reply：OK, many thanks!

If you have any further question, please feel free to contact me. 

Sincerely,

Dunming Xiao,

On behalf of all authors.

---

## [Decision Letter · Decision Letter 1]

2 Sep 2024

Long Term Cost-effectiveness Analysis of IDegLira in the Treatment of Type 2 Diabetes Patients Compared to GLP-1RA Added to Basal Insulin after IDegLira Entered the National Reimbursement Drug List in China

PONE-D-24-13585R1

Dear Dr. Chen,

We’re pleased to inform you that your manuscript has been judged scientifically suitable for publication and will be formally accepted for publication once it meets all outstanding technical requirements.

Kind regards,

Timotius Ivan Hariyanto, M.D.

Academic Editor

PLOS ONE

Additional Editor Comments (optional):

Reviewers' comments:

Reviewer's Responses to Questions

**Comments to the Author**

1. If the authors have adequately addressed your comments raised in a previous round of review and you feel that this manuscript is now acceptable for publication, you may indicate that here to bypass the “Comments to the Author” section, enter your conflict of interest statement in the “Confidential to Editor” section, and submit your "Accept" recommendation.

Reviewer #1: All comments have been addressed

Reviewer #2: All comments have been addressed

2. Is the manuscript technically sound, and do the data support the conclusions?

Reviewer #1: Yes

Reviewer #2: Yes

3. Has the statistical analysis been performed appropriately and rigorously? 

Reviewer #1: Yes

Reviewer #2: Yes

4. Have the authors made all data underlying the findings in their manuscript fully available?

Reviewer #1: Yes

Reviewer #2: Yes

5. Is the manuscript presented in an intelligible fashion and written in standard English?

Reviewer #1: Yes

Reviewer #2: Yes

6. Review Comments to the Author

Reviewer #1: (No Response)

Reviewer #2: (No Response)

7. PLOS authors have the option to publish the peer review history of their article (what does this mean?). If published, this will include your full peer review and any attached files.

Reviewer #1: No

Reviewer #2: No

---

## [Editor Report · Acceptance letter]

15 Sep 2024

PONE-D-24-13585R1 

PLOS ONE

Dear Dr. Chen, 

I'm pleased to inform you that your manuscript has been deemed suitable for publication in PLOS ONE. Congratulations! Your manuscript is now being handed over to our production team.

Kind regards, 

on behalf of

Dr. Timotius Ivan Hariyanto 

Academic Editor

PLOS ONE